# Vitamin D and Beta Cells in Type 1 Diabetes: A Systematic Review

**DOI:** 10.3390/ijms232214434

**Published:** 2022-11-20

**Authors:** Josephine Yu, Preeti Sharma, Christian M. Girgis, Jenny E. Gunton

**Affiliations:** 1Centre for Diabetes, Obesity and Endocrinology (CDOE), The Westmead Institute for Medical Research, The University of Sydney, Westmead, Sydney, NSW 2145, Australia; 2Faculty of Medicine and Health, The University of Sydney, Camperdown, Sydney, NSW 2050, Australia; 3Department of Diabetes and Endocrinology, Westmead Hospital, Westmead, Sydney, NSW 2145, Australia

**Keywords:** vitamin D, type 1 diabetes, vitamin D receptor, beta-cell, insulin, C-peptide

## Abstract

The prevalence of type 1 diabetes (T1D) is rising steadily. A potential contributor to the rise is vitamin D. In this systematic review, we examined the literature around vitamin D and T1D. We identified 22 papers examining the role of vitamin D in cultured β-cell lines, islets, or perfused pancreas, and 28 papers examining vitamin D in humans or human islets. The literature reports strong associations between T1D and low circulating vitamin D. There is also high-level (systematic reviews, meta-analyses) evidence that adequate vitamin D status in early life reduces T1D risk. Several animal studies, particularly in NOD mice, show harm from D-deficiency and benefit in most studies from vitamin D treatment/supplementation. Short-term streptozotocin studies show a β-cell survival effect with supplementation. Human studies report associations between VDR polymorphisms and T1D risk and β-cell function, as assessed by C-peptide. In view of those outcomes, the variable results in human trials are generally disappointing. Most studies using 1,25D, the active form of vitamin D were ineffective. Similarly, studies using other forms of vitamin D were predominantly ineffective. However, it is interesting to note that all but one of the studies testing 25D reported benefit. Together, this suggests that maintenance of optimal circulating 25D levels may reduce the risk of T1D and that it may have potential for benefits in delaying the development of absolute or near-absolute C-peptide deficiency. Given the near-complete loss of β-cells by the time of clinical diagnosis, vitamin D is much less likely to be useful after disease-onset. However, given the very low toxicity of 25D, and the known benefits of preservation of C-peptide positivity for long-term complications risk, we recommend considering daily cholecalciferol supplementation in people with T1D and people at high risk of T1D, especially if they have vitamin D insufficiency.

## 1. Introduction

The prevalence of type 1 diabetes (T1D) has been rising steadily over a number of decades in most countries with regular reporting [1]. One potential contributor to the rising rate of T1D is vitamin D.

Vitamin D is not truly a vitamin (a vital substance which cannot be synthesised in humans) as it is synthesised in humans from 7-dehydrocholesterol. Upon UV exposure to the skin, 7-dehydrocholesterol is converted to vitamin D3. This undergoes 2 successive hydroxylation steps to form 25(OH)vitamin D and then 1,25(OH)_2_vitamin D, which we will abbreviate to 25D and 1,25D, respectively.

The 25-hydroxylation step primarily occurs in the liver and 25D is the predominant circulating form of vitamin D. In people with optimal vitamin D status, 25D is present at 75–150 nmol/L (30–60 ng/mL) in the circulation. Levels of <50 nmol/L are consistent with vitamin D deficiency. There is a feedback interaction between 25D and parathyroid hormone (PTH) levels, with the nadir of PTH release at 25D concentrations around 75–80 nmol/L. The second hydroxylation step to make 1,25D from 25D is carried out by the enzyme 1α-hydroxylase, which is encoded by the gene CYP27B1. 1,25D is the active hormone, which binds to the vitamin D receptor (VDR), which is a transcription factor. The enzyme 1α-hydroxylase is highly expressed in kidney, and renal activation of 1,25D accounts for most of the circulating 1,25D [2]. However, other cell types can also produce circulating 1,25D, including macrophages and placenta [2,3]. In addition, many tissues possess capacity to produce local 1,25D including muscle and pancreatic islets.

Vitamin D in early life modulates the developing immune system, and it is important in the normal development and maintenance of self-tolerance [4,5,6,7,8,9,10]. Impaired vitamin D signalling, especially in early life, increases the risk of autoimmunity [8,11,12]. Human studies show that adequate vitamin D, especially in early life, protects against the development of T1D [4,10]. Vitamin D deficiency is more common in people with T1D [13,14,15], including newly diagnosed patients [16].

However, even in the setting of autoimmunity, T1D does not occur until there is substantial death of pancreatic β-cells. The role for vitamin D in β-cell function, survival and conversion from autoimmunity to T1D is less well understood and is the subject of this systematic review (search shown in Figure 1).

## 2. Results and Discussion

### 2.1. Cell Culture Studies

Vitamin D receptor is one of the most highly expressed transcription factors in mouse β-cells [17]. Human protein atlas reports quite high *VDR* mRNA expression in pancreatic endocrine cells https://www.proteinatlas.org/ENSG00000111424-VDR/single+cell+type (accessed on 26 June 2022).

Using an iPS cell line differentiated into β-cell-like cells, Wei et al. found that knockdown of VDR caused increased sensitivity to cytokine-induced β-cell death [17]. From the same paper, INS-1 cells with VDR knockdown had impaired gene expression after challenge with cytokines and palmitate [17]. Knockdown of VDR by siRNA in human EndoC-βH1 cells decreased cell count by >50% at 36 h with associated increase in reactive oxygen species (ROS) and increased Foxo1 [18].

Riachy et al. used the RINm5F rat β-cell line to study effects of 1,25D at what was referred to as ‘physiological’ (10^−8^ M) and ‘pharmacological’ (10^−6^ M) doses [19]. Since 1,25D circulates at picomolar concentrations, both doses are high. Treatment improved maintenance of normal cell function with combined cytokine treatment (combined interferon-γ (IFNγ), interleukin-1β (IL1β) and tumour necrosis factor α (TNFα)). Vitamin D also decreased apoptosis with cytokine treatment without notable increase in benefit with the very high dose 1,25D.

In INS-1e cells and isolated rat β-cells, Gysemans reported that 1,25D reduced cytokine-overexpression but did not find a reduction in cytokine-induced β-cell death [20].

Pepaj examined INS-1 cells and used SILAC (stable isotope labelling by amino acids in cell culture) and mass spectrometry to examine proteins induced by vitamin D treatment [21]. They found a 7-fold up-regulation of Tmem27, demonstrating that vitamin D regulates protein production in β-cells. Tmem27 increases β-cell proliferation [22]. Functional studies in β-cells were not reported.

Using MIN6 cells, Hu reported that 1,25D (5 nM, pre-treatment) improved endoplasmic reticulum (ER) stress and reduced apoptosis induced by H_2_O_2_ [23]. In other studies of MIN6 cells, vitamin D (10 pM 1,25D) improved anti-apoptotic gene expression and reduced β-cell apoptosis [24].

Lee et al. used RIN 1046-38 cells and did not find any increase in insulin release following treatment with high doses of 1,25D (50–100 nM for 48 or 72 h). They found that 1,25D inhibited cell growth to 69% of control levels [25]. This inhibition of cell growth is consistent with that seen in other cell types such as C2C12 myocytes [26].

### 2.2. Islet Studies

Human islets from 5 donors were studied by Riachy et al. and treated with 10^−8^ M or 10^−6^ M 1,25D. They found induction of the anti-apoptotic protein A20 following treatment with 1,25D and a decrease in apoptosis by Hoechst [19]. In a follow-up study they reported that 1,25D altered Fas induction and performed microarrays, showing many regulated genes in cell death pathways [27].

As well as the cell studies described above, Wei also studied isolated islets treated with the vitamin D analogue calcipotriol and found that this protected them against IL-1β-induced cell death [17].

Sandler et al. used rat islets treated ± IL-1β and either 1,25D or vitamin D analogues (MC903 or KH1060), each of which improved islet insulin release [28]. In these rat islets, 1,25D was beneficial down to 100 pM concentrations.

Mendes et al. studied a dexamethasone-induced insulin resistance model to investigate the effects of vitamin D on insulin resistance and β-cells. They found improved insulin secretion in islets and associated changes in calcium-handling [29]. While this is a type 2 diabetes-like model, it demonstrates clear effects of vitamin D on β-cell function.

### 2.3. Animal Studies

#### 2.3.1. Non-Obese Diabetic (NOD) Mice

Using a vitamin D analogue Ro 26-2198, Gregon et al. showed decreased progression to T1D in NOD mice with improved β-cell survival [30]. There was a significant effect to decrease insulitis/islet infiltration. Additionally, using NOD mice, van Etten et al. found that a high dose of 1,25D (25 µg/d) or analogues, TX522 or TX527, reduced the incidence of T1D [31] and improved pancreatic insulin content. In NOD mice lacking the Ins2 gene, a vitamin D analogue (2α-methyl-19-nor-(20S)-1,25vitaminD3) also decreased progression to T1D [32] with decreased insulitis. In these studies there were no reports of effects on isolated islets, so while the studies demonstrate β-cell survival, they are not able to separate immunomodulatory effects from direct β-cell effects.

Gysemans studied female NOD mice treated with 5 µg/kg/2 days of 1,25D and again found decreased insulitis and diabetes risk [20]. After 14 weeks of age, when insulitis is usually established, 1,25D did not alter diabetes development. They also examined islets from NOD-SCID mice, which are immunodeficient and therefore do not develop insulitis. Vitamin D treatment reduced chemokine expression in the islets, but did not alter β-cell death.

Using diabetic NOD mice receiving syngeneic islet transplants from non-diabetic NOD mice, Baeke et al. [33] found that the analogue TX527 substantially improved efficacy of αCD3-antibody and αCD3+cyclosporin to prevent transplant rejection. Direct β-cell studies were not reported.

Low doses of anti-CD3, cyclosporin A and the vitamin D analogue, TX527, synergise to delay recurrence of autoimmune diabetes in an islet-transplanted NOD mouse model of diabetes [33].

NOD mice treated with intra-peritoneal vitamin D (2200 IU/kg for 28 days) exhibited improved blood glucose and insulin secretion [34].

#### 2.3.2. Other Models

Vitamin D receptor overexpression was used by Morro et al. to test susceptibility to streptozotocin-induced diabetes. They found that VDR-transgenic mice were resistant to streptozotocin and had better preservation of β-cell mass following administration of the toxin [35]. Transgenic-VDR mice also had elevated markers of β-cell proliferation after streptozotocin. *Vdr* was nutrient-regulated; with higher glucose in tissue culture and fed-state in vivo both associated with greater *Vdr* expression [35].

Using Wistar rats with diabetes induced by streptozotocin or by high-fat diet plus low-dose streptozotocin, Sadek et al. found 10 IU/kg of vitamin D given 3 days after diabetes induction caused significant improvement in β-cell function and HbA1c [36]. Pancreatic β-cell mass was not reported.

VDR-null mice on normal diet develop hypocalcaemia, and accompanying β-cell dysfunction [37].In that circumstance, VDR-NOD mice have accelerated development of diabetes [37] but this is ameliorated when the mice are fed a rescue diet with high calcium, lactulose and phosphate. We have previously reported that rescue diet in VDR-null mice normalises serum calcium and phosphate levels [38].

Rats fed a vitamin D deficient diet had markedly reduced insulin secretion from perfused pancreas compared to pair-fed control-diet rats [39,40]. In the deficient rats, insulin secretion was reduced in response to glucose, arginine and the sulfonylurea tolbutamide [39,40].

In C57Bl/6 mice treated with multiple low-dose streptozotocin, vitamin D (1,25D, 5 µg/kg/2 days commencing 1 h before 1st dose of streptozotocin) improved blood glucose levels and serum insulin, reducing diabetes incidence from 100% to 40% [24].

Mice with deletion of D binding protein (DBP), which carries circulating forms of vitamin D have a greater number of smaller islet α-cells and impaired glucagon release. This is associated with lower blood glucose levels and improved insulin sensitivity [41]. β-cell survival and type 1 diabetes models were not examined in mice. In the same study, they analysed human pancreata and found that α-cells are DBP+ on histology, in both normal glucose tolerant individuals and in people with diabetes [41].

#### 2.3.3. Negative Mouse Studies

Hawa et al. treated NOD/Ba mice with 16 IU daily of vitamin D from mothers conception to 15 days post-natally and then in offspring until 10 weeks of age. No difference in diabetes was found, nor in degree of insulitis or pancreatic insulin content [42]. As mentioned above, when NOD mice were treated after 14 weeks of age, vitamin D was ineffective [20].

In contrast to results of Driver et al. [37], Gysemans et al. [43] also studied VDR-null NOD mice and reported no effects of VDR deletion upon incidence of diabetes in mice fed normal chow. The reasons for the different results between the 2 studies are unclear.

#### 2.3.4. Conclusions; Animal Studies

Overall, most animal studies show that vitamin D deficiency increases risk of T1D and conversely that treatment with vitamin D decreases risk. Very few are able to differentiate the effects on β-cells from other cells, except for short-term studies examining β-cell toxins such as streptozotocin. Longer term studies would not exclude an immunological effect, but diabetes-induction within a week following streptozotocin is unlikely to be due to triggered autoimmunity. The short term studies all report benefit of increased vitamin D [24,35,36].

### 2.4. Human Studies

#### 2.4.1. Human Polymorphism Studies (Table 1)

Many studies have examined whether polymorphisms in VDR are associated with diabetes, and this was the subject of a paper by Zhang et al. which meta-analysed 26 studies of gene polymorphisms in T1D [44]. They found no effect of the *Fok*I site, but a significant association with *Bsm*I bb genotype and T1D (Odds ratio (OR) 1.30). When considering population subgroups, *Bsm*I associated with greater risk in Asian populations (OR 2.15) and was not different from OR of 1 in other groups. No effect was seen in the meta-analysis of *Apa*I or *Taq*I polymorphisms.

Studies not incorporated in that meta-analysis include Rasoul and colleagues who examined 4 VDR gene polymorphisms in 253 T1D and 214 control Kuwaiti subjects. They reported significant differences at rs10735810 *Fok*I C > T and rs731236 *Taq*I C > T with both being more common with T1D [45]. In contrast 189 people with T1D and 194 controls, genotyping rs10735810 and rs154410, there were no associations with auto-antibodies in a Brazilian population [46]. There were no significant associations between the polymorphisms and measures of residual β-cell function although the f-allele of rs10735810 tended towards association with impaired β-cell function.

**Table 1 ijms-23-14434-t001:** Single nucleotide polymorphisms of vitamin D receptor gene and potential functions.

Ref	Gene Polymorphism	SNP	Mutation	Location	Effect
[44]	*Fok*I	rs2228570	C > T	Exon 2	Translation start site influences VDR activity
[45,47]	*Fok*I	rs10735810	C > T	Exon 2	Translation start site influences VDR activity
[44]	*Taq*I	rs731236	C > T	3′ UTR	Modulation of mRNA stability
[44]	*Bsm*I	rs1544410	A > G	3′ UTR	Modulation of mRNA stability
[44]	*Apa*I	rs7975232	A > C	3′ UTR	Modulation of mRNA stability

In an Iranian study, 101 children with T1D had VDR polymorphisms examined and compared to residual β-cell function. The study reported several polymorphisms in VDR to be associated with greater fasting and/or stimulated C-peptide [48]. Serum vitamin D levels were also positively associated with higher C-peptide.

Overall, it appears that *Bsm*I may be important in Asian populations for T1D risk, and potentially there may be a role for *Fok*I polymorphisms in some groups.

#### 2.4.2. Human Observational and Cross-Sectional Studies and Case Reports

Most studies find that circulating vitamin D levels are lower in people with type 1 diabetes [13,14,15,49]. A study of nutritional factors in T1D reported that serum 25D was unexpectedly inversely correlated with residual C-peptide [50]. At face value, this suggests higher vitamin D is associated with impaired β-cell mass or function. However, this relationship could alternately be a marker of improved insulin sensitivity with higher vitamin D [51]; less insulin (and thus C-peptide) is required with better insulin sensitivity. Examination of stimulated rather than fasting C-peptide could potentially clarify the possible insulin sensitivity versus β-cell function.

Not all studies find this association [52] with no difference in 25D or DBP in one small study of 36 people with T1D and 27 controls. Examining autoimmunity, another study found no association between vitamin D status and β-cell auto-antibodies or T-cell FOXP3 expression [53].

Breast-feeding is associated with reduced risk of T1D and lack of vitamin D supplementation in early life is associated with increased risk [54]. Adequate early life vitamin D status, especially in the first 6 months of life, decreases T1D risk, and conversely, clinically suspected rickets is associated with increased T1D risk [55]. Fronczak et al. reported increased maternal vitamin D intake in food decreased risk of islet autoimmunity in their offspring No effect of the 1alpha,25-dihydroxyvitamin D3 on beta-cell residual function and insulin requirement in adults [56]. However, in another study of maternal vitamin D intake, there was no association between dietary or supplemental intake in the mothers and offspring autoimmunity or diabetes [57]. A case–control study which was part of EURODIAB found that vitamin D supplementation was associated with decreased risk of T1D diagnosis before 15 years of age (OR 0.63) [58].

Early life supplementation with cod liver oil, which contains high amounts of vitamin D, decreases T1D in childhood [59]. It is not possible to exclude benefits from other components in cod-liver oil.

Kodama and colleagues found autoantibodies to DBP in a greater proportion of people with T1D than controls [60]. They also found that DBP was expressed in human pancreas, predominantly in α-cells.

As discussed above, 1,25D treatment of isolated human islets reduces cell death in response to cytokines [19,27].

Baidal and colleagues reported a single case with newly diagnosed T1D treated with a combination of high-dose omega-3-fatty acids and 1,25D, showed improvements in C-peptide at 9 and 12 months [61].

Of course, none of these studies are able to confirm or refute separate effects of vitamin D upon β-cells from other, immunological effects.

#### 2.4.3. Therapeutic Trials (Table 2)

Mishra et al. describe a case–control intervention study of supplementing children with 2000 IU/day of 25D in 15 children with recent onset T1D [62] who were compared to 15 controls. They report a trend towards slower decrease in C-peptide, but the differences were not statistically significant.

An open-label observational study of calciferol 2000 IU/d plus etanercept weekly during the first 3 months plus GAD-alum injections on days 30 and 60 was carried out in 20 T1D patients aged 12 years [63].

Pitocco et al. conducted an open-label RCT of 70 newly diagnosed people who were given 0.25 µg 2nd-daily of 1,25D versus 25 mg/kg nicotinamide. The study was powered to detect approximately a doubling of C-peptide at 1 year, equating to complete preservation of baseline C-peptide. There was no difference in C-peptide at 1 year, although patients receiving 1,25D had significantly lower insulin requirements at 3 and 6 months [64].

Gabbay and colleagues [65] conducted a placebo-controlled RCT in 35 slightly older patients (7–30 years) with onset up to 6 months ago. The group had normal mean baseline vitamin D (65 nmol/L), and the baseline HbA1c was 1.5% higher in the D-group (*p* < 0.05). Participants were treated with 2000 IU of 25D daily or matching placebo. Significantly fewer subjects in the D-group progressed to low C-peptide 19% versus 62%.

Examining vitamin D in established diabetes, Sharma et al. tested the effect of 60000 IU of 25D monthly in 52 children with diabetes for a mean of >4 years [66]. In the D-group, C-peptide increased at the time of the primary endpoint (6 months) and decreased in the controls, which was significant.

Islet autoimmunity was examined in a study of 15 patients, of whom 8 were treated with 25D to target circulating levels >125 nmol/L [67]. They found significantly decreased ELISpot activity against GAD, IA-2 and pro-insulin in the treated group. Insulin doses also decreased in the D-group, from 0.39 to 0.3 U/kg/day without significant difference in C-peptide or HbA1c.

Alfacalcidol at 0.25 µg bd was used in a single-blind RCT of 61 children with new-onset T1D [68]. It had no significant effects. However, post hoc analysis found lower insulin dose and enhanced C-peptide response in boys, without significant benefit in girls.

Using ergocalciferol (vitamin D2) at 50000 IU per week for 2 months, Nwosu et al. reported a slower decline in HbA1c in the active treatment group [69]. At baseline, the controls had significantly lower insulin-dose-adjusted HbA1c, which the authors interpreted as consistent with better baseline β-cell function in the controls. That benefit was lost over the trial, such that the 2 groups finished with similar insulin-dose-adjusted HbA1c.

Li et al. used 1α(OH)vitamin D3 at 0.5 µg/day in an RCT of 35 people and reported improved fasting C-peptide in the treatment group. However, the D-group started ~50% higher than the controls (not significant) and the 2 groups finished the study with near-identical levels [70].

Using 1,25D at 0.25 µg/day in 27 adults with recent onset T1D, Napoli et al. found no effects on β-cell function, or, in post hoc analyses upon markers of bone turnover [60]. In a 2-year RCT, Bizzarri et al. tested 25 µg of 1,25D in 34 people, of whom almost half were D-deficient at baseline. There were no significant effects at 12 or 24 months except higher vitamin D [71].

**Table 2 ijms-23-14434-t002:** Human therapeutic trials of vitamin D and analogues.

1st Author	Ref	Design	N (Loss)	Age	Treatment	Notes and Effects	Strengths	Weaknesses
Pitocco	[64]	Open-label RCT T1D < 4 w	70 (3)	>5 y	1,25D 0.25 µg 2nd daily or nicotinamide	↓ insulin dose at 3 and 6 m	Moderate size RCT. Low loss of subjects	Large age range
Gabbay	[65]	RCT. T1D < 6 m	35	7–30 y	25D 2000 IU daily vs. placebo	Basal D 65 nmol/L. 19% vs. 62% progression to low C-peptide	RCT (PC, DB)	↑ basal HbA1c in D (9.2 vs. 7.7%).
Federico	[67]	8 deficient people given D	15	12 ± 0.9 y	25D to achieve serum 25D > 125 nmol/L	Decreased immunoreactivity to GAD, IA2 and Pro-insulin. ↓ insulin dose with D.	Individual dosing to target 25D	Small n
Ataie-Jafari	[68]	Single-blind RCT. T1D < 8 w	61 (7)	8–15 y	Alfacalcidol 0.25 µg bd	No significant effects. Post hoc males ↓ insulin dose, ↑ C-peptide	Small age range. RCT	Assessed FCP Short duration
Bizzarri	[71]	RCT. T1D < 12 w FCP > 0.25 nmol/L	34 (7)	11–35 y	0.25 µg 1,25D daily	2 y follow-up. No effects.	RCT (PC, DB), long follow-up	Medium effect size due to dropout
Li	[70]	RCT, blinding unclear. LADA < 5 y FCP > 0.2 nmol/L	35	38.5 D, 42.8 con	1α(OH)vitamin D3 0.5 µg/d	FCP ↓ in controls (started ~50% higher), stable in D group. Similar FCP at end.	High compliance records (daily)	Small n Controls ↑ basal FCP
Napoli	[60]	RCT. T1D < 12 w	27	22 y	1,25D 0.25 µg/d	No significant effects	Controlled RCT	Small n
Sharma	[66]	RCT. Duration T1D 4.75 y and 4 y.	52	9–9.5 y	25D, 60,000 IU monthly	↑ FCP with D, vs. ↓ in controls. No change in insulin or Hba1c.	Baseline values similar	Small n
Nwosu	[69]	RCT. T1D < 3 m	36	10–21 y	Ergocalciferol (D2) 2 m of 50,000 IU/w	Slower rise in HbA1c.	RCT (PC, DB) Long follow-up	Single centre and small n
Walter	[72]	RCT	40	18–39 y	1,25D 0.25 µg/d	No differences, decline in both groups.	Low dropout rate	Small n
Mishra	[62]	Open label case–control	30	6–12 y	25D. 2000 IU/d	No significant differences. Trend to slower decline in D group.	Age-matched patients	Small n and relatively short term
Panjiyar	[73]	Open label case–control	72	6–12 y	25D. 3000 IU/d in 42 patients. 30 controls	Smaller decline in C-peptide, ↓ HbA1c	Sufficient n with long follow-up	Imbalanced control vs. treatment groups
Ludvig-sson	[63]	Open label	20	12.4	Calciferol 2000 IU/d + etanercept + GAD-alum	Higher C-peptide at 6 m. No obvious long-term benefits	Long follow-up	Small n

N (loss) = number in study and (loss to follow-up). DB, PC = double-blind, placebo-controlled. FCP = fasting C-peptide. ↑ = increased ↓ = decreased. A Brazilian group compared 7 subjects who received adipose stem cell transplant + 25D 2000 IU daily to 2 controls [74]. The active group had a ~40% increase in C-peptide which was not seen in the controls.

Walter et al. also used 1,25D, at 0.25 µg/day and found no benefit in 40 adults with T1D [72]. Both groups experienced similar decline in β-cell function over the time of the study.

Panjiyar et al. also reported a case–control study of 72 subjects, 42 of whom were treated with 25D at 3000 IU/d. They found a significantly smaller decline in C-peptide in the D-treated group and better HbA1c [73].

An open-label observational study of calciferol 2000 IU/d plus etanercept weekly during the first 3 months plus GAD-alum injections on days 30 and 60 was carried out in 20 T1D patients aged 12 years [63]. While C-peptide increased at 6 months, there were no obvious long-term protective effects.

## 3. Methods and Materials

PubMed was searched using the terms (((type 1 diabetes) OR (insulin dependent diabetes)) OR (juvenile onset diabetes)) AND (vitamin D) AND (beta-cell) (Figure 1). Only papers written in English were included. This search identified 119 papers. From these we identified primary data papers examining the role of vitamin D in β-cells in T1D. There were 22 papers which examined the role of vitamin D, predominantly in cultured β-cell lines, islets, or perfused pancreas [17,19,20,21,23,24,25,27,28,29,30,31,32,33,34,35,36,37,39,40,41,42], and 28 papers examined vitamin D in humans or human islets [15,19,27,45,46,48,50,52,53,54,56,57,58,60,61,62,63,64,65,66,67,68,69,70,71,72,73,74,75,76]. Studies which examined the effect on immune cells without diabetes or β-cell phenotype are not discussed further in this review. The remaining papers were reviews or did not examine effects on β-cells.

## 4. Conclusions

There are strong associations between T1D and low circulating vitamin D. There is also high-level (systematic reviews, meta-analyses) evidence that adequate vitamin D status in early life reduces the risk of diabetes [77,78].

Several animal studies, particularly in NOD mice, show harm from D-deficiency and benefit in most studies from vitamin D treatment/supplementation. Short-term streptozotocin studies show a β-cell survival effect with supplementation studies. Human studies report associations between VDR polymorphisms and T1D risk and β-cell function, as assessed by C-peptide. These findings have been summarized below in Table 3.

In view of these outcomes, the variable results in human trials are generally disappointing. It is possible that these differences arise from various factors including, but not limited to, compound tested (25D vs. 1,25D), duration of intervention, concentration, and onset of disease. However, it is interesting to note that all but one of the studies testing 25D alone (Table 2) reported beneficial effects [65,66,67,69,73] with the ineffective study not finding significant benefit but reporting a trend to benefit [62]. Most studies using 1,25D, the active form of vitamin D were ineffective [60,71,72], with only one open-label study in 70 subjects [64] showing lower insulin dose only at 3 and 6 months. Similar to the studies using 1,25D, studies using other forms of vitamin D were also predominantly ineffective [63,68,70] although the Nwosu study found a slower rise in HbA1c [69].

As discussed in the introduction, 1,25D is the active ligand for VDR. However, it is 25D that is the predominant form in circulation where it is found in nanomolar concentrations, compared to picomolar levels for 1,25D. It has relatively recently been demonstrated that pancreatic β-cells are able to convert 25D to 1,25D [17]. This, and the above trial data suggests that exogenous and circulating 1,25D may have little effect on β-cell survival in T1D. However, boosting 25D levels would provide more substrate for local β-cell formation of 1,25D which may then have autocrine and / or paracrine effects.

Together, this suggests that maintenance of optimal circulating 25D levels is important to reduce the risk of T1D and that it may have potential for benefits in delaying the development of absolute or near-absolute C-peptide deficiency. Given the near-complete loss of β-cells by the time of clinical diagnosis, vitamin D is much less likely to be useful after disease-onset. However, given the very low toxicity of 25D treatment, and the known benefits of preservation of C-peptide positivity for long-term complications risk, we recommend considering ≥2000 IU of daily cholecalciferol supplementation in people with T1D and people at high risk of T1D, especially if they have low 25D levels. We recommend cholecalciferol due to the reduced risk of hypercalcemia and relatively long half-life. Future studies should consider assessing the risk of T1D development and progression with and without at least 2000 IU of cholecalciferol.

## Figures and Tables

**Figure 1 ijms-23-14434-f001:**
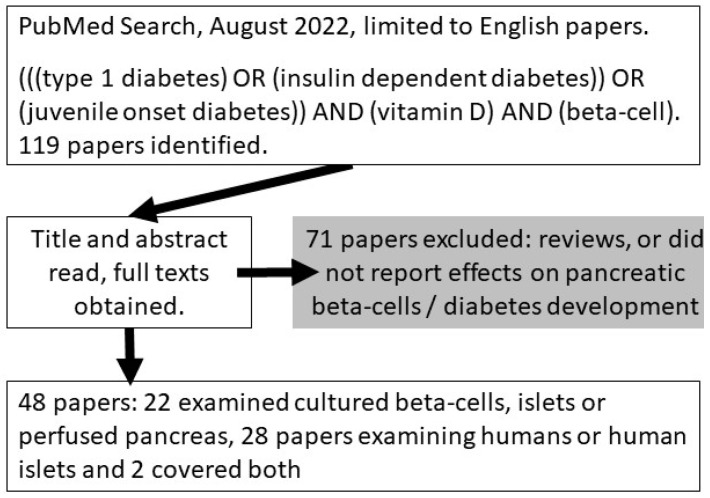
Flow chart of literature review and included/excluded papers.

**Table 3 ijms-23-14434-t003:** Summary of study outcomes.

Model	Treatments	Primary Outcomes
Cell/Cell Culture Studies		
β-cell lines	1,25D	Improved maintenance of normal cell function with cytokines [19] Increased β-cell proliferation [22] Reduced ER stress and apoptosis in the presence of H_2_O_2_ [23] Effects on cell survival unclear: pro-survival [19,24], no effect [20]
Isolated rat β-cells	1,25D	Reduced cytokine overexpression, no change cellular apoptosis when treated with cytokines [20] Improved islet insulin release [28]
Human islets	1,25D	Reduced apoptosis [19,27]
Human IPS β-like cells	Calcipotriol	Reduced apoptosis in presence of IL-1β [17]
Animal Studies		
NOD mice	Analogues	Improved β-cell survival and decreased insulitis [30] Reduced incidence of T1D and improved pancreatic insulin [20,31,32] Decreased progression to T1D (mice lacking *Ins2* gene) [32] Improved efficacy of islet transplants and delayed recurrence of autoimmune diabetes [33]
	1,25D	Reduced incidence of T1D and improved pancreatic insulin [31]
VDR mice	Transgenic	Preservation of β-cell mass following STZ [35]
	Null	Develop hypocalcaemia and β-cell dysfunction [37] which is ameliorated by rescue diet (high calcium, lactulose, phosphate) [38]
C57Bl/6 mice	1,25D	Improved blood glucose and serum insulin when treated with low-dose STZ [24]
Diabetic Wistar rats	Vitamin D	Improved β-cell function and HbA1c [36]
Human Studies		
Early life	Supplementation with vitamin D	Reduced risk of T1D development intake of vitamin D and risk of type 1 diabetes: a birth-cohort study [55,59] Conflicting evidence regarding increased maternal vitamin D intake (pro No effect of the 1alpha,25-dihydroxyvitamin D3 on beta-cell residual function and insulin requirement in adults vs. no effect [58]
Recent onset T1D	25D	Beneficial effects on C-peptide levels over time (trend to slower decrease in C-peptide [62], if not elevated C-peptide levels [66]
	Analogues	Lower insulin dose and enhanced C-peptide (boys only) [68]
Recent onset T1D	1,25D	Lower insulin requirements at 3 and 6 months [64]
	25D	Enhanced/higher C-peptide levels [65,73,74] Lower insulin requirements [67] Improved HbA1c [73]
	Analogues	Slower decline in HbA1c levels over time [69]
Human Studies (Negative/Null Findings)		
Recent onset T1D	Combined Rx + D	Study in children. No beneficial effects [63]
Recent onset T1D	1,25D	No effects on β-cell function [60,71,72]

## Data Availability

Not applicable.

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
