# Peer review of "Vitamin D and Beta Cells in Type 1 Diabetes: A Systematic Review"

_ijms, 2022, doi:10.3390/ijms232214434_

Round 1

Reviewer 1 Report

Manuscript iD: ijms-1994337

Authors: Yu et al.,

In this systemic review entitled “Vitamin D and Beta Cells in Type 1 Diabetes: A Systematic Review”, the authors studied the role for vitamin D in β-cell function, survival and conversion from autoimmunity to type 1 diabetes (T1D) by analyzing the available 22 papers on the role of vitamin D in cultured β-cell lines, islets, or perfused pancreas, and 28 other papers, which focus on vitamin D in humans or human islets.

Hereafter, some points that should be taken into account before processing further.

Comments to the authors:

-          It would be better to use “inclusion chart” in the title of figure 2, as “chart for identification” sounds misleading.

-          Number and/or font of the subtitles for the different sections and subsections should be different, as all of them are similar in the current version of the manuscript.

-          The authors should add the corresponding location of rs10735810 (Taql gene polymorphism), which is “Exon 2”.

-          The borders of table 1 should be checked.

-          The effect of C>T mutation in rs10735810 should be precized rather than “Receptor protein structure”, which is general.

-          English language is fine. Just small checking/editing is required.

-           

Author Response

Reviewer 1

In this systemic review entitled “Vitamin D and Beta Cells in Type 1 Diabetes: A Systematic Review”, the authors studied the role for vitamin D in β-cell function, survival and conversion from autoimmunity to type 1 diabetes (T1D) by analyzing the available 22 papers on the role of vitamin D in cultured β-cell lines, islets, or perfused pancreas, and 28 other papers, which focus on vitamin D in humans or human islets.
Hereafter, some points that should be taken into account before processing further. Comments to the authors:

It would be better to use “inclusion chart” in the title of figure 2, as “chart for identification” sounds misleading.

Response: Thank you, we have changed the legend to
Figure 1. Flow chart of literature review and included/excluded papers.

Number and/or font of the subtitles for the different sections and subsections should be different, as all of them are similar in the current version of the manuscript.

Response: We thank the reviewer for their comment. We have now modified the formatting to allow the reader to readily distinguish the subtitles and different subsections.
Major Headings: RESULTS
Subheadings: Animal studies
Subsections under subheadings: Non-obese diabetic (NOD) mice

The authors should add the corresponding location of rs10735810 (Taql gene polymorphism), which is “Exon 2”. The borders of table 1 should be checked. The effect of C>T mutation in rs10735810 should be precized rather than “Receptor protein structure”, which is general.

Response: We thank the reviewer for their comments. We have specified the location for rs10735810 (FokI gene polymorphism) and added the borders for table 1 (shown below), such that it remains consistent with the formatting for table 2. We have also modified the effect of C>T mutation in rs10735810 as suggested by the reviewer.

English language is fine. Just small checking/editing is required.

Response: We thank the reviewer for their comments. The author team have checked and made some corrections (tracked).

Reviewer 2 Report

This is a good review of a very relevant topic involving the role of vitamin D as a potential contributor to the development of type 1 diabetes.

One methodological issue in this study is that a better understanding should be provided to the readers on how the literature reviewed was graded, i.e., what is the quality of the evidence? This  information is particularly important to the human therapeutic trials summarized in table 2. That table should be expanded to include strengths and weaknesses of each human therapeutic trial. Given the recommendation provided by the authors toward considering the use of 2,000 IU of daily calciferol supplementation to those with or at high risk of type 1 diabetes, authors should enhance the discussion in the conclusions section to incorporate additional ideas/recommendations on the design of future studies that can determine the effectiveness of their proposed therapeutic approach.   

A table illustrating an overall summary by study categories could facilitate visualization of the status of the research/findings and what could be next steps. Similarly, further discussion might be needed on potential reasons behind animal studies showing increased risk of type 1 diabetes with  vitamin deficiency and a decrease in such risk  with vitamin D treatment. Why are clinical trials conducted so far unable to replicate these findings in humans? And what could be done to enhance translation of findings into clinical practice?

This is a good review of a very relevant topic involving the role of vitamin D as a potential contributor to the development of type 1 diabetes.

One methodological issue in this study is that a better understanding should be provided to the readers on how the literature reviewed was graded, i.e., what is the quality of the evidence? This  information is particularly important to the human therapeutic trials summarized in table 2. That table should be expanded to include strengths and weaknesses of each human therapeutic trial. Given the recommendation provided by the authors toward considering the use of 2,000 IU of daily calciferol supplementation to those with or at high risk of type 1 diabetes, authors should enhance the discussion in the conclusions section to incorporate additional ideas/recommendations on the design of future studies that can determine the effectiveness of their proposed therapeutic approach.   

A table illustrating an overall summary by study categories could facilitate visualization of the status of the research/findings and what could be next steps. Similarly, further discussion might be needed on potential reasons behind animal studies showing increased risk of type 1 diabetes with  vitamin deficiency and a decrease in such risk  with vitamin D treatment. Why are clinical trials conducted so far unable to replicate these findings in humans? And what could be done to enhance translation of findings into clinical practice?

This is a good review of a very relevant topic involving the role of vitamin D as a potential contributor to the development of type 1 diabetes.

One methodological issue in this study is that a better understanding should be provided to the readers on how the literature reviewed was graded, i.e., what is the quality of the evidence? This  information is particularly important to the human therapeutic trials summarized in table 2. That table should be expanded to include strengths and weaknesses of each human therapeutic trial. Given the recommendation provided by the authors toward considering the use of 2,000 IU of daily calciferol supplementation to those with or at high risk of type 1 diabetes, authors should enhance the discussion in the conclusions section to incorporate additional ideas/recommendations on the design of future studies that can determine the effectiveness of their proposed therapeutic approach.   

A table illustrating an overall summary by study categories could facilitate visualization of the status of the research/findings and what could be next steps. Similarly, further discussion might be needed on potential reasons behind animal studies showing increased risk of type 1 diabetes with  vitamin deficiency and a decrease in such risk with vitamin D treatment. Why are clinical trials conducted so far unable to replicate these findings in humans? And what could be done to enhance translation of findings into clinical practice?

This is a good review of a very relevant topic involving the role of vitamin D as a potential contributor to the development of type 1 diabetes.

One methodological issue in this study is that a better understanding should be provided to the readers on how the literature reviewed was graded, i.e., what is the quality of the evidence? This  information is particularly important to the human therapeutic trials summarized in table 2. That table should be expanded to include strengths and weaknesses of each human therapeutic trial. Given the recommendation provided by the authors toward considering the use of 2,000 IU of daily calciferol supplementation to those with or at high risk of type 1 diabetes, authors should enhance the discussion in the conclusions section to incorporate additional ideas/recommendations on the design of future studies that can determine the effectiveness of their proposed therapeutic approach.   

A table illustrating an overall summary by study categories could facilitate visualization of the status of the research/findings and what could be next steps. Similarly, further discussion might be needed on potential reasons behind animal studies showing increased risk of type 1 diabetes with  vitamin deficiency and a decrease in such risk  with vitamin D treatment. Why are clinical trials conducted so far unable to replicate these findings in humans? And what could be done to enhance translation of findings into clinical practice?

This is a good review of a very relevant topic involving the role of vitamin D as a potential contributor to the development of type 1 diabetes.

One methodological issue in this study is that a better understanding should be provided to the readers on how the literature reviewed was graded, i.e., what is the quality of the evidence? This  information is particularly important to the human therapeutic trials summarized in table 2. That table should be expanded to include strengths and weaknesses of each human therapeutic trial. Given the recommendation provided by the authors toward considering the use of 2,000 IU of daily calciferol supplementation to those with or at high risk of type 1 diabetes, authors should enhance the discussion in the conclusions section to incorporate additional ideas/recommendations on the design of future studies that can determine the effectiveness of their proposed therapeutic approach.   

A table illustrating an overall summary by study categories could facilitate visualization of the status of the research/findings and what could be next steps. Similarly, further discussion might be needed on potential reasons behind animal studies showing increased risk of type 1 diabetes with  vitamin deficiency and a decrease in such risk  with vitamin D treatment. Why are clinical trials conducted so far unable to replicate these findings in humans? And what could be done to enhance translation of findings into clinical practice?

This is a good review of a very relevant topic involving the role of vitamin D as a potential contributor to the development of type 1 diabetes.

One methodological issue in this study is that a better understanding should be provided to the readers on how the literature reviewed was graded, i.e., what is the quality of the evidence? This  information is particularly important to the human therapeutic trials summarized in table 2. That table should be expanded to include strengths and weaknesses of each human therapeutic trial. Given the recommendation provided by the authors toward considering the use of 2,000 IU of daily calciferol supplementation to those with or at high risk of type 1 diabetes, authors should enhance the discussion in the conclusions section to incorporate additional ideas/recommendations on the design of future studies that can determine the effectiveness of their proposed therapeutic approach.   

A table illustrating an overall summary by study categories could facilitate visualization of the status of the research/findings and what could be next steps. Similarly, further discussion might be needed on potential reasons behind animal studies showing increased risk of type 1 diabetes with  vitamin deficiency and a decrease in such risk  with vitamin D treatment. Why are clinical trials conducted so far unable to replicate these findings in humans? And what could be done to enhance translation of findings into clinical practice?

This is a good review of a very relevant topic involving the role of vitamin D as a potential contributor to the development of type 1 diabetes.

One methodological issue in this study is that a better understanding should be provided to the readers on how the literature reviewed was graded, i.e., what is the quality of the evidence? This  information is particularly important to the human therapeutic trials summarized in table 2. That table should be expanded to include strengths and weaknesses of each human therapeutic trial. Given the recommendation provided by the authors toward considering the use of 2,000 IU of daily calciferol supplementation to those with or at high risk of type 1 diabetes, authors should enhance the discussion in the conclusions section to incorporate additional ideas/recommendations on the design of future studies that can determine the effectiveness of their proposed therapeutic approach.   

A table illustrating an overall summary by study categories could facilitate visualization of the status of the research/findings and what could be next steps. Similarly, further discussion might be needed on potential reasons behind animal studies showing increased risk of type 1 diabetes with  vitamin deficiency and a decrease in such risk  with vitamin D treatment. Why are clinical trials conducted so far unable to replicate these findings in humans? And what could be done to enhance translation of findings into clinical practice?

This is a good review of a very relevant topic involving the role of vitamin D as a potential contributor to the development of type 1 diabetes.

One methodological issue in this study is that a better understanding should be provided to the readers on how the literature reviewed was graded, i.e., what is the quality of the evidence? This  information is particularly important to the human therapeutic trials summarized in table 2. That table should be expanded to include strengths and weaknesses of each human therapeutic trial. Given the recommendation provided by the authors toward considering the use of 2,000 IU of daily calciferol supplementation to those with or at high risk of type 1 diabetes, authors should enhance the discussion in the conclusions section to incorporate additional ideas/recommendations on the design of future studies that can determine the effectiveness of their proposed therapeutic approach.   

A table illustrating an overall summary by study categories could facilitate visualization of the status of the research/findings and what could be next steps. Similarly, further discussion might be needed on potential reasons behind animal studies showing increased risk of type 1 diabetes with  vitamin deficiency and a decrease in such risk  with vitamin D treatment. Why are clinical trials conducted so far unable to replicate these findings in humans? And what could be done to enhance translation of findings into clinical practice?

This is a good review of a very relevant topic involving the role of vitamin D as a potential contributor to the development of type 1 diabetes.

One methodological issue in this study is that a better understanding should be provided to the readers on how the literature reviewed was graded, i.e., what is the quality of the evidence? This  information is particularly important to the human therapeutic trials summarized in table 2. That table should be expanded to include strengths and weaknesses of each human therapeutic trial. Given the recommendation provided by the authors toward considering the use of 2,000 IU of daily calciferol supplementation to those with or at high risk of type 1 diabetes, authors should enhance the discussion in the conclusions section to incorporate additional ideas/recommendations on the design of future studies that can determine the effectiveness of their proposed therapeutic approach.   

A table illustrating an overall summary by study categories could facilitate visualization of the status of the research/findings and what could be next steps. Similarly, further discussion might be needed on potential reasons behind animal studies showing increased risk of type 1 diabetes with  vitamin deficiency and a decrease in such risk  with vitamin D treatment. Why are clinical trials conducted so far unable to replicate these findings in humans? And what could be done to enhance translation of findings into clinical practice?

This is a good review of a very relevant topic involving the role of vitamin D as a potential contributor to the development of type 1 diabetes.

One methodological issue in this study is that a better understanding should be provided to the readers on how the literature reviewed was graded, i.e., what is the quality of the evidence? This  information is particularly important to the human therapeutic trials summarized in table 2. That table should be expanded to include strengths and weaknesses of each human therapeutic trial. Given the recommendation provided by the authors toward considering the use of 2,000 IU of daily calciferol supplementation to those with or at high risk of type 1 diabetes, authors should enhance the discussion in the conclusions section to incorporate additional ideas/recommendations on the design of future studies that can determine the effectiveness of their proposed therapeutic approach.   

A table illustrating an overall summary by study categories could facilitate visualization of the status of the research/findings and what could be next steps. Similarly, further discussion might be needed on potential reasons behind animal studies showing increased risk of type 1 diabetes with  vitamin deficiency and a decrease in such risk  with vitamin D treatment. Why are clinical trials conducted so far unable to replicate these findings in humans? And what could be done to enhance translation of findings into clinical practice?

This is a good review of a very relevant topic involving the role of vitamin D as a potential contributor to the development of type 1 diabetes.

One methodological issue in this study is that a better understanding should be provided to the readers on how the literature reviewed was graded, i.e., what is the quality of the evidence? This  information is particularly important to the human therapeutic trials summarized in table 2. That table should be expanded to include strengths and weaknesses of each human therapeutic trial. Given the recommendation provided by the authors toward considering the use of 2,000 IU of daily calciferol supplementation to those with or at high risk of type 1 diabetes, authors should enhance the discussion in the conclusions section to incorporate additional ideas/recommendations on the design of future studies that can determine the effectiveness of their proposed therapeutic approach.   

A table illustrating an overall summary by study categories could facilitate visualization of the status of the research/findings and what could be next steps. Similarly, further discussion might be needed on potential reasons behind animal studies showing increased risk of type 1 diabetes with  vitamin deficiency and a decrease in such risk  with vitamin D treatment. Why are clinical trials conducted so far unable to replicate these findings in humans? And what could be done to enhance translation of findings into clinical practice?

This is a good review of a very relevant topic involving the role of vitamin D as a potential contributor to the development of type 1 diabetes.

One methodological issue in this study is that a better understanding should be provided to the readers on how the literature reviewed was graded, i.e., what is the quality of the evidence? This  information is particularly important to the human therapeutic trials summarized in table 2. That table should be expanded to include strengths and weaknesses of each human therapeutic trial. Given the recommendation provided by the authors toward considering the use of 2,000 IU of daily calciferol supplementation to those with or at high risk of type 1 diabetes, authors should enhance the discussion in the conclusions section to incorporate additional ideas/recommendations on the design of future studies that can determine the effectiveness of their proposed therapeutic approach.   

A table illustrating an overall summary by study categories could facilitate visualization of the status of the research/findings and what could be next steps. Similarly, further discussion might be needed on potential reasons behind animal studies showing increased risk of type 1 diabetes with  vitamin deficiency and a decrease in such risk  with vitamin D treatment. Why are clinical trials conducted so far unable to replicate these findings in humans? And what could be done to enhance translation of findings into clinical practice?

This is a good review of a very relevant topic involving the role of vitamin D as a potential contributor to the development of type 1 diabetes.

One methodological issue in this study is that a better understanding should be provided to the readers on how the literature reviewed was graded, i.e., what is the quality of the evidence? This  information is particularly important to the human therapeutic trials summarized in table 2. That table should be expanded to include strengths and weaknesses of each human therapeutic trial. Given the recommendation provided by the authors toward considering the use of 2,000 IU of daily calciferol supplementation to those with or at high risk of type 1 diabetes, authors should enhance the discussion in the conclusions section to incorporate additional ideas/recommendations on the design of future studies that can determine the effectiveness of their proposed therapeutic approach.   

A table illustrating an overall summary by study categories could facilitate visualization of the status of the research/findings and what could be next steps. Similarly, further discussion might be needed on potential reasons behind animal studies showing increased risk of type 1 diabetes with  vitamin deficiency and a decrease in such risk  with vitamin D treatment. Why are clinical trials conducted so far unable to replicate these findings in humans? And what could be done to enhance translation of findings into clinical practice?

This is a good review of a very relevant topic involving the role of vitamin D as a potential contributor to the development of type 1 diabetes.

One methodological issue in this study is that a better understanding should be provided to the readers on how the literature reviewed was graded, i.e., what is the quality of the evidence? This  information is particularly important to the human therapeutic trials summarized in table 2. That table should be expanded to include strengths and weaknesses of each human therapeutic trial. Given the recommendation provided by the authors toward considering the use of 2,000 IU of daily calciferol supplementation to those with or at high risk of type 1 diabetes, authors should enhance the discussion in the conclusions section to incorporate additional ideas/recommendations on the design of future studies that can determine the effectiveness of their proposed therapeutic approach.   

A table illustrating an overall summary by study categories could facilitate visualization of the status of the research/findings and what could be next steps. Similarly, further discussion might be needed on potential reasons behind animal studies showing increased risk of type 1 diabetes with  vitamin deficiency and a decrease in such risk  with vitamin D treatment. Why are clinical trials conducted so far unable to replicate these findings in humans? And what could be done to enhance translation of findings into clinical practice?

This is a good review of a very relevant topic involving the role of vitamin D as a potential contributor to the development of type 1 diabetes.

One methodological issue in this study is that a better understanding should be provided to the readers on how the literature reviewed was graded, i.e., what is the quality of the evidence? This  information is particularly important to the human therapeutic trials summarized in table 2. That table should be expanded to include strengths and weaknesses of each human therapeutic trial. Given the recommendation provided by the authors toward considering the use of 2,000 IU of daily calciferol supplementation to those with or at high risk of type 1 diabetes, authors should enhance the discussion in the conclusions section to incorporate additional ideas/recommendations on the design of future studies that can determine the effectiveness of their proposed therapeutic approach.   

A table illustrating an overall summary by study categories could facilitate visualization of the status of the research/findings and what could be next steps. Similarly, further discussion might be needed on potential reasons behind animal studies showing increased risk of type 1 diabetes with  vitamin deficiency and a decrease in such risk  with vitamin D treatment. Why are clinical trials conducted so far unable to replicate these findings in humans? And what could be done to enhance translation of findings into clinical practice?

This is a good review of a very relevant topic involving the role of vitamin D as a potential contributor to the development of type 1 diabetes.

One methodological issue in this study is that a better understanding should be provided to the readers on how the literature reviewed was graded, i.e., what is the quality of the evidence? This  information is particularly important to the human therapeutic trials summarized in table 2. That table should be expanded to include strengths and weaknesses of each human therapeutic trial. Given the recommendation provided by the authors toward considering the use of 2,000 IU of daily calciferol supplementation to those with or at high risk of type 1 diabetes, authors should enhance the discussion in the conclusions section to incorporate additional ideas/recommendations on the design of future studies that can determine the effectiveness of their proposed therapeutic approach.   

A table illustrating an overall summary by study categories could facilitate visualization of the status of the research/findings and what could be next steps. Similarly, further discussion might be needed on potential reasons behind animal studies showing increased risk of type 1 diabetes with  vitamin deficiency and a decrease in such risk  with vitamin D treatment. Why are clinical trials conducted so far unable to replicate these findings in humans? And what could be done to enhance translation of findings into clinical practice?

This is a good review of a very relevant topic involving the role of vitamin D as a potential contributor to the development of type 1 diabetes.

One methodological issue in this study is that a better understanding should be provided to the readers on how the literature reviewed was graded, i.e., what is the quality of the evidence? This  information is particularly important to the human therapeutic trials summarized in table 2. That table should be expanded to include strengths and weaknesses of each human therapeutic trial. Given the recommendation provided by the authors toward considering the use of 2,000 IU of daily calciferol supplementation to those with or at high risk of type 1 diabetes, authors should enhance the discussion in the conclusions section to incorporate additional ideas/recommendations on the design of future studies that can determine the effectiveness of their proposed therapeutic approach.   

A table illustrating an overall summary by study categories could facilitate visualization of the status of the research/findings and what could be next steps. Similarly, further discussion might be needed on potential reasons behind animal studies showing increased risk of type 1 diabetes with  vitamin deficiency and a decrease in such risk  with vitamin D treatment. Why are clinical trials conducted so far unable to replicate these findings in humans? And what could be done to enhance translation of findings into clinical practice?

This is a good review of a very relevant topic involving the role of vitamin D as a potential contributor to the development of type 1 diabetes.

One methodological issue in this study is that a better understanding should be provided to the readers on how the literature reviewed was graded, i.e., what is the quality of the evidence? This  information is particularly important to the human therapeutic trials summarized in table 2. That table should be expanded to include strengths and weaknesses of each human therapeutic trial. Given the recommendation provided by the authors toward considering the use of 2,000 IU of daily calciferol supplementation to those with or at high risk of type 1 diabetes, authors should enhance the discussion in the conclusions section to incorporate additional ideas/recommendations on the design of future studies that can determine the effectiveness of their proposed therapeutic approach.   

A table illustrating an overall summary by study categories could facilitate visualization of the status of the research/findings and what could be next steps. Similarly, further discussion might be needed on potential reasons behind animal studies showing increased risk of type 1 diabetes with  vitamin deficiency and a decrease in such risk  with vitamin D treatment. Why are clinical trials conducted so far unable to replicate these findings in humans? And what could be done to enhance translation of findings into clinical practice?

Author Response

This is a good review of a very relevant topic involving the role of vitamin D as a potential contributor to the development of type 1 diabetes.

One methodological issue in this study is that a better understanding should be provided to the readers on how the literature reviewed was graded, i.e., what is the quality of the evidence? This information is particularly important to the human therapeutic trials summarized in table 2. That table should be expanded to include strengths and weaknesses of each human therapeutic trial.

Response: As per the reviewer’s suggestion, we have now integrated two additional columns to table 2, to address the strengths and weaknesses of each therapeutic trial as shown below.

Given the recommendation provided by the authors toward considering the use of ≥ 2,000 IU of daily calciferol supplementation to those with or at high risk of type 1 diabetes, authors should enhance the discussion in the conclusions section to incorporate additional ideas/recommendations on the design of future studies that can determine the effectiveness of their proposed therapeutic approach.  

Response: We have taken this suggestion onboard and have integrated a future studies section at the end of our conclusion.

However, given the very low toxicity of 25D treatment, and the known benefits of preservation of C-peptide positivity for long-term complications risk, we recommend considering ≥2000 IU of daily cholecalciferol supplementation in people with T1D and people at high risk of T1D, especially if they have low 25D levels. We recommend cholecalciferol due to the reduced risk of hypercalcemia and relatively long half-life. Future studies should consider assessing the risk of T1D development and progression with and without at least 2000 IU of cholecalciferol.

A table illustrating an overall summary by study categories could facilitate visualization of the status of the research/findings and what could be next steps.

Response: Thank you, we have created a new Table 3

Several animal studies, particularly in NOD mice, show harm from D-deficiency and benefit in most studies from vitamin D treatment/supplementation. Short-term streptozotocin studies show a β-cell survival effect with supplementation studies. Human studies report associations between VDR polymorphisms and T1D risk and β-cell function, as assessed by C-peptide. These findings have been summarized below in Table 3.

Similarly, further discussion might be needed on potential reasons behind animal studies showing increased risk of type 1 diabetes with vitamin deficiency and a decrease in such risk with vitamin D treatment. Why are clinical trials conducted so far unable to replicate these findings in humans? And what could be done to enhance translation of findings into clinical practice?

Response: We understand the concerns brought up by the reviewer and have modified our conclusion section to elaborate on the matter. The human studies using cholecalciferol are generally consistent with the animal studies.

For future potential trial design, cholecalciferol appears to be the best agent, and studies should assess  baseline vitamin D status and achieved vitamin D levels, using sufficient cholecalciferol to achieve normal 25D status.

Round 2

Reviewer 1 Report

Report for ijms-1994337

After checking the second version of the manuscript, we can obviously notice that it has been improved following the comments addressed by the referees. I think the article warrant publication as I have no further comments.

Reviewer 2 Report

Authors revised the manuscript accordingly. It looks great now with the additional information provided.  Ready for publication.